# A Study on the Factors Influencing the Intention and Behavior Deviation of Rural Residents in Waste Separation—Based on LOGIT-ISM-MICMAC Combination Model

**Xue-Yuan Li [1]**, **Sen-Wei Huang [1,\*]**, **Qian Lin [2]**, **Qiu-Jia Lu [3]** and **Ya-Shan Zhang [1]**

1   School of Public Management, Fujian Agriculture and Forestry University, Fuzhou 350002, China
2   School of Foreign Languages, Huaqiao University, Quanzhou 362000, China
3   School of Economics and Management, Fujian Agriculture and Forestry University, Fuzhou 350002, China
\*   Correspondence: hsw@fafu.edu.cn

**Abstract:** Accurate identification of the influencing factors and the mechanisms of the willingness-behavior paradox in rural residents in waste separation is conducive to stimulating rural residents to participate in rural environmental governance, which is important for solving the willingness-behavior paradox problem. By using CLES data, we analyzed the factors influencing rural residents' willingness to separate garbage and behavioral paradoxes using the combined LOGIT-ISM-MICMAC model. The results of the study showed that (1) the regression results showed that eight factors, including publicity means, reward and punishment means, policy effect perception, villagers' environmental protection behavior perception, gender, age, socio-economic status, and ecological livability status, affect the paradox of villagers' willingness to separate garbage and behavior; (2) the results of the ISM model show that there are four main transmission paths, and the commonality lies in the common transmission paths of "policy publicity effect factor, villagers' perception of environmental protection behavior, village ecological habitability, and deviation of willingness and behavior"; (3) the results of MICMAC model show that we should focus on strengthening the ecological habitat of villages, ensuring the effectiveness of policy promotion, and encouraging villagers' environmental protection behavior to reduce the deviation of rural residents' behavior and intention.

**Keywords:** waste separation; behavior; willingness; paradox; rural environment

## 1. Introduction

Ecological livability is one of the main goals of the rural revitalization strategy. To achieve the goal of ecological livability in rural areas, it is not only necessary to speed up the treatment of rural sewage and rural toilet revolution, but also the treatment of rural domestic garbage is an important part of achieving the goal of ecological livability. In China's Five-Year Action Plan for the Improvement and Enhancement of Rural Habitat Environment (2021–2025), it is also mentioned that "we should accelerate the source classification and reduction in rural domestic garbage, actively explore classification and treatment modes that meet the characteristics of rural areas and farmers' habits and are simple and easy to implement, and reduce the amount of garbage disposed of out of villages." It is urgent that we explore the sustainable treatment mode of rural garbage. Currently, there are a variety of rural domestic waste treatment models in China, but the different rural domestic waste treatment model's governance efficiency is not much improved [1]. How to stimulate the vitality and improve the efficiency of rural household waste treatment models? It has been pointed out that the efficiency of rural household waste treatment models requires strengthening the regulatory role of the government and the participation of rural residents [2]. However, the overall level of participation of rural residents is generally low, and there is a discrepancy between willingness to participate and participation behavior, mainly showing the characteristics of "high willingness-low

behavior" [3]. In this regard, it is of practical and theoretical significance to study how to promote the transformation from willingness to behavior and to study the reasons and influencing factors of the deviation between willingness and participation behavior to promote the efficiency of rural household waste management model, which is also the key to promote rural household waste management.

To study the reasons for the deviation of willingness and behavior, it is important to understand what factors influence willingness and behavior. Social behavior and planned behavior theories suggest that individuals have a strong consistency between their intentions and behaviors. So why do intentions and behaviors diverge? Studies have been conducted to investigate the factors influencing willingness and behavior and willingness-behavior paradoxes in terms of different categories of willingness and behavior and willingness-behavior paradoxes, mainly in terms of three cognitive dimensions based on the theory of planned behavior [4,5], autonomous and embedded factors under the social embedding theory [6], two dimensions of external and internal environment [7], and individual characteristics and family characteristics to investigate the influence of willingness and behavior and willingness-behavior paradoxes factors. Among them, studies on waste separation in developing countries show that household economic and social welfare [8], the amount and distance of infrastructure for waste disposal [9], and neighborhood factors [10] are important factors influencing willingness and behavior to separate waste. For developed countries, empirical studies in Taiwan and Japan have shown that incentives [11] and coercive measures [12] are important factors influencing the willingness and behavior of waste separation. There are also studies by examining the factors influencing household recycling behavior in Western European countries and comparing them with household recycling behavior in the U.S. The results show that socioeconomic factors and incentive policies affect household recycling behavior, and household recycling behavior in Western European countries is more influenced by incentive policies, and although Americans' recycling behavior is also influenced by policy factors [13], compared with Western European countries, U.S. household recycling behavior is more influenced by socioeconomic conditions and infrastructure than in Western European countries [14].

The existing studies have provided important references for analyzing the causal factors and influencing factors of willingness and behavioral deviance of rural residents in waste sorting, but the following issues still need to be further explored: (1) scholars have studied the causes of willingness and behavioral deviance of urban residents in waste sorting, but there is a lack of research on willingness and behavioral deviance of rural residents in waste sorting. (2) Among the research dimensions based on existing studies, although the theory of social embeddedness incorporates the cognitive factors of the theory of planned behavior and considers the influencing factors of external environment, it still lacks the exploration of the factors of village characteristics. Therefore, we need to enrich and improve the theory of social embeddedness. (3) At present, relevant studies often use the logit-ISM combination model to analyze the influencing factors of willingness-behavior paradox and the research path of influencing factors, but they lack the in-depth discussion of the importance of influencing factors. Based on this, the logit-ISM model is further deepened to the LOGIT-ISM-MICMAC model to investigate the strength of the influence factors on the overall system. Based on the research dimensions of existing research and considering the availability of data, we will fully incorporate the influence factors and add variables of village characteristics so as to further improve the study of rural residents' willingness to sort garbage and behavior paradox.

## 2. Research Methods, Data Sources and Variable Selection

### 2.1. Research Methodology

The reason for using the combined LOGIT-ISM-MICMAC model is that the traditional econometric model can identify the causal relationship between the influencing factors, and the ISM model can further investigate the transmission path between the influencing factors with a causal relationship. The MICMAC model can identify the strength of the

influencing factors based on the results of the ISM model and can further investigate the importance of the influencing factors in the overall system and propose countermeasures and suggestions in a more targeted manner.

### 2.1.1. Logit Model

Logit model can solve the problem of non-normality of independent variables and is suitable for non-linear situation. The dependent variable in the paper is whether rural residents' willingness to participate in waste separation is contrary to their behavior, which is a dichotomous variable, so the logit model is chosen for analysis, and the specific model is as follows.

$$Y_i = ln\left(\frac{P_i}{1 - P_i}\right) = \beta_0 + \beta_1 X_1 + \beta_2 X_2 + \ldots + \beta_n X_n + \varepsilon \tag{1}$$

In the Equation (1), it denotes the paradox of rural residents' willingness to separate garbage from their behavior, where the probability of $Y_i$ taking 1 is $P_i$ and the probability of $Y_i$ taking 0 is $1 - P_i$. The $Y_i$ denotes the paradoxical situation between the willingness and behavior of rural residents to separate garbage, and $X_n$ $(n = 1, 2, \ldots n)$ denotes the possible factors that may influence the deviation between the willingness and behavior of rural residents to separate garbage, and $\beta_n (n = 1, 2, \ldots, n)$ denotes the regression coefficient of the nth independent variable, and $\varepsilon$ denotes the random error term.

### 2.1.2. ISM Model

The Interpretative Structural Modeling (ISM) Model is usually used to study the structure and hierarchical relationship of the elements within the system [15] based on the combination of logit regression results and the experience of experts and scholars; the influence factors with a large number of variables and complex structural relationships are stratified and graded as a way to explore in depth the conduction path of the influence factors and the hierarchy of each factor in the system of rural residents' willingness to separate waste and behavior paradox. The specific operation is to identify the causal relationship between each influencing factor and form the adjacency matrix, then calculate the reachable matrix according to Boolean operation and obtain the conduction path of influencing factors through the hierarchical decomposition of the reachable matrix [16].

The adjacency matrix needs to be derived from the logical relationship between the influencing factors. The logical relationship between the influencing factors is whether any two influencing factors will affect each other. The components of the adjacency matrix A are defined as follows:

$$a_{ij} = \begin{cases} 1 & S_i \text{ and } S_j \text{ are related} \\ 0 & S_i \text{ and } S_j \text{ are not related} \end{cases} \tag{2}$$

where $i = 1, 2, \cdots, n$, the $j = 1, 2, \cdots, n$.

With the help of Boolean operators, the adjacency matrix A can be transformed into a reachable matrix M, I is the unit matrix, which is calculated as follows:

$$M = (A + I)^{\lambda+1} = (A + I)^{\lambda} \neq (A + I)^{\lambda-1} \tag{3}$$

The hierarchical elements from the top to the bottom of the hierarchy are determined in the following manner:

$$L = \{S_i | P(S_i) \cap Q(S_i) = C(S_i)\} \tag{4}$$

where $P(S_i)$ is the reachable set, which is the set consisting of all the columns with matrix element 1 in the row corresponding to the influencing factor $S_i$ in the reachable matrix M. $Q(S_i)$ is the prior set, which is the set consisting of all the rows with matrix element 1 in the column corresponding to the influencing element $S_i$ in the reachable matrix M, whose

intersection is defined as $C(S_i)$. All the elements in $P(S_i)$ corresponding to the influencing factor are top-level elements when and only when $P(S_i) \cap Q(S_i)$, and then the top-level elements are eliminated. Continue this step until the bottom level.

### 2.1.3. MICMAC Model

Based on the results of hierarchical division of influencing factors by the ISM model, the MICMAC analysis method is used to make a deep analysis of the position and role of influencing factors by calculating the dependency and drive of each influencing factor and to propose corresponding countermeasures and suggestions. On the basis of obtaining the reachable matrix M, MICMAC analysis is conducted. Dependency is the number of elements corresponding to the column in which each factor in M is located as 1, and drive is the number of elements corresponding to the row in which each factor in M is located as 1. And from this, all indicators can be divided into five regions: I (autonomous factor), II (dependent factor), III (associated factor), IV (driver), and V (adjustment factor) [17]. The specific formulae for calculating dependence and drive are as follows:

$$E_j = \sum_{i=1}^{n} m_{ij}(j = 1, 2, \cdots, n) \tag{5}$$

$$F_i = \sum_{j=1}^{n} m_{ij}(i = 1, 2, \cdots, n) \tag{6}$$

In Equations (5) and (6), $E_j$ is the dependency; $F_i$ is the driving force, the $m_{ij}$ is denoted as the influence factor in the reachable matrix M.

### 2.2. Data Sources

The China Land Economic Survey (CLES) was founded by the Humanities and Social Sciences Division of Nanjing Agricultural University in 2020, and the Jinshanbao Institute of Agricultural Modernization assisted in implementing the survey. The team established and surveyed fixed observation sites in rural Jiangsu Province to comprehensively analyze the current situation of rural, social, and economic development in Jiangsu. The survey questionnaire covers land market, agricultural production, rural industry, ecological environment, poverty alleviation, and rural finance. The survey adopts PPS sampling, and 26 research districts and counties are selected among 13 prefecture-level cities in the Jiangsu Province, 2 sample townships are selected in each district and county, 1 administrative village is selected in each township, and 50 farming households are randomly selected in each village. The total sample was 52 administrative villages and 2600 farming households. According to the purpose of the study and the screening of the questionnaires, the questionnaire samples with missing values were excluded, and finally, 2204 valid questionnaire samples were obtained.

### 2.3. Variable Selection

Based on the purpose of the study and related research literature, the difference between willingness to participate and participation behavior was selected as the dependent variable, and the remaining possible influencing factors were divided into five categories: individual characteristics, family characteristics, village characteristics, external environment, and internal factors, with a total of 22 variables. Among them, "Do you know about rural habitat improvement?" and "Do you know about rural living environment improvement?" were selected as the policy effect perceptions. and "Do you know about rural household waste classification?" Two questions were obtained by entropy method (see Table 1 for details).

**Table 1.** Descriptive statistics table.

| | Variable Name | Meaning and Assignment | Average Value | Standard Deviation |
|---|---|---|---|---|
| Dependent variable | Deviation of willful behavior | Do intentions and behaviors contradict each other? 0 = contradictory; 1 = no contradiction. Use the questions "Are you willing to sort your household waste? (1 = yes; 0 = no)" and "Do you separate your household waste? (1 = yes; 0 = no)", the groups with willingness without behavior and without willingness with behavior were defined as contradictory, and the groups with willingness with behavior and without willingness without behavior were defined as non-contradictory. | 0.530 | 0.499 |
| Individual Characteristics | Gender | 0 = Female; 1 = Male | 0.714 | 0.452 |
| | Age | Continuous variables (weeks of age) | 60.655 | 11.376 |
| | Education level | Continuous variables (years) | 7.164 | 3.829 |
| | Health Status | 1 = incapacitated; 2 = poor; 3 = moderate; 4 = good; 5 = excellent | 3.948 | 1.057 |
| | Socio-economic status | How do you feel about your local economic status? 1 = Very low; 2 = Low; 3 = Ordinary; 4 = High; 5 = Very high | 2.936 | 0.707 |
| Family Characteristics | Resident population | How many people live in your household (6 months of the year or more)? (people) | 3.236 | 1.651 |
| | Availability of cadres | 0 = none; 1 = yes | 0.156 | 0.363 |
| | Whether there are party members | 0 = none; 1 = yes | 0.309 | 0.462 |
| | Are you religious | 0 = none; 1 = yes | 0.062 | 0.241 |
| Village Features | Industrial prosperity | Satisfaction with the prosperity of industries in this village (industrial layout, vitality of industrial development, driving employment of rural residents, etc.). 1 = very dissatisfied; 2 = less satisfied; 3 = fair; 4 = more satisfied; 5 = very satisfied | 3.446 | 0.967 |
| | Ecological Livability | Satisfaction with the ecological livability of the village (village appearance, living convenience, sewage and garbage management, air quality, etc.). 1 = very dissatisfied; 2 = less satisfied; 3 = fair; 4 = more satisfied; 5 = very satisfied | 4.109 | 0.733 |
| | Countryside Civilization | Satisfaction with the village's rural culture and civilization (rural ideological and moral construction, quality of compulsory education, quality of services of the village integrated cultural service center, etc.). 1 = very dissatisfied; 2 = less satisfied; 3 = fairly satisfied; 4 = more satisfied; 5 = very satisfied | 4.019 | 0.730 |
| | Effective governance | Satisfaction with the effectiveness of governance in this village (village leadership, security management in the village, openness of village affairs, etc.). 1 = very dissatisfied; 2 = less satisfied; 3 = fair; 4 = more satisfied; 5 = very satisfied | 4.080 | 0.751 |
| External Environment | Promotional Tools | Has the government publicized the separation of rural household waste? 0 = No; 1 = Yes | 0.868 | 0.338 |
| | Reward and punishment means | Regarding the separation of rural household waste, has the government implemented incentives and penalties? 0 = No; 1 = Yes | 0.211 | 0.408 |
| Internal factors | Importance Perception | Do you agree that the separation of domestic waste has a positive effect on the improvement of the rural environment? 1 = Don't agree at all; 2 = Don't agree very much; 3 = Generally agree; 4 = Like to agree; 5 = Completely agree | 4.312 | 0.935 |

**Table 1.** *Cont.*

| Variable Name | Meaning and Assignment | Average Value | Standard Deviation |
|---|---|---|---|
| Policy Effect Perception | Entropy method of finding values, continuous variables, the higher the value the stronger the perception | 3.006 | 1.060 |
| Perceived environmental behavior of villagers | Your attitude towards other villagers' environmental behavior? 1 = disagree; 2 = fair; 3 = strongly agree | 2.500 | 0.531 |
| Blood Trust | Level of trust in relatives? 1 = very distrustful; 2 = relatively distrustful; 3 = fair; 4 = relatively trusting; 5 = relatively trusting | 4.271 | 0.775 |
| Geographic Trust | Level of trust in neighbors? 1 = very distrustful; 2 = relatively distrustful; 3 = average; 4 = relatively trustful; 5 = very trustful | 4.009 | 0.772 |
| Cadre Trust | Level of trust in village officials? 1 = very distrustful; 2 = relatively distrustful; 3 = fair; 4 = relatively trusting; 5 = relatively trusting | 4.056 | 0.803 |
| Social Networks | Number of your mobile contacts (people) | 101.546 | 450.202 |

## 3. Results and Analysis

*3.1. Analysis of the Factors Influencing the Paradox of Rural Residents' Willingness and Behavior to Separate Garbage*

A binary Logit model was established for the factors influencing the paradoxical behavior and willingness of rural residents to separate garbage, and 2204 data were processed using Stata. Meanwhile, to ensure the robustness of the model results, the Probit model was used to replace the Logit model for robustness testing, and the results of the two models remained consistent overall, proving the robustness of the Logit model results (the estimation results of the Logit model are detailed in Table 2).

**Table 2.** Model regression results.

| Variable Category | Variable Name | Coefficient | Standard Deviation | $p$-Value |
|---|---|---|---|---|
| Individual Characteristics | Gender | −0.197 | 0.115 | 0.086 * |
| | Age | −0.019 | 0.005 | 0.000 *** |
| | Education level | 0.008 | 0.015 | 0.604 |
| | Health Status | −0.026 | 0.049 | 0.599 |
| | Socio-economic status | 0.212 | 0.070 | 0.003 *** |
| Family Characteristics | Resident population | −0.036 | 0.029 | 0.225 |
| | Availability of cadres | −0.155 | 0.142 | 0.278 |
| | Whether there are party members | −0.026 | 0.113 | 0.821 |
| | Are you religious? | −0.065 | 0.203 | 0.749 |
| Village Features | Industrial prosperity | 0.019 | 0.054 | 0.725 |
| | Ecological Livability | 0.248 | 0.083 | 0.003 *** |
| | Countryside Civilization | −0.128 | 0.088 | 0.144 |
| | Effective governance | 0.008 | 0.081 | 0.924 |
| External Environment | Promotional Tools | 1.048 | 0.162 | 0.000 *** |
| | Reward and punishment means | 1.161 | 0.131 | 0.000 *** |
| | Importance Perception | −0.020 | 0.053 | 0.702 |
| | Policy Effect Perception | 0.362 | 0.053 | 0.000 *** |
| | Perceived environmental behavior of villagers | 0.503 | 0.092 | 0.000 *** |
| Internal factors | Blood Trust | −0.086 | 0.083 | 0.297 |
| | Geographic Trust | −0.023 | 0.086 | 0.786 |
| | Cadre Trust | 0.043 | 0.077 | 0.578 |
| | Social Networks | 0.000 | 0.000 | 0.273 |
| | Constant term | −2.685 | 0.591 | 0.000 *** |
| | $R^2$ | | 0.138 | |

Note: (1) *, *** indicate that each variable is significant at the 10%, and 1% levels, respectively.

(1) Individual characteristics. The gender variable significantly affects the deviation of rural residents' willingness to separate garbage and behavior at the 10% level, indicating that there is a gender difference in the deviation of rural residents' willingness to separate garbage and behavior, and the degree of deviation is higher for males than females, which may be explained by the fact that males are less involved in household activities and females are mainly responsible for household activities, thus leading to the gender difference in the deviation of behavior. The age variable negatively affects the deviation between the willingness and behavior of rural residents at the 1% level, indicating that the older the rural residents are, the higher the deviation between their willingness and behavior, which may be explained by the fact that the older the rural residents tend to be retired or retired from work and have more free time for waste separation in general, but their knowledge or understanding of waste separation is insufficient. The lack of knowledge or understanding of waste separation leads to a deviation in their willingness and behavior. Socio-economic status also positively influenced the deviation between the willingness and behavior of rural residents at the 1% level, indicating that rural residents with better economic conditions have lower deviation between their willingness and behavior, indicating they are more willing to participate in waste separation activities.

(2) Village characteristics. The ecological livability variable positively and significantly affects the deviation between rural residents' willingness and behavior of waste separation at the 1% level, indicating that the better the village characteristics, such as village appearance, living convenience, sewage and waste management, and air quality, the lower the deviation between rural residents' willingness and behavior of waste separation, and the more the village attaches importance to the construction of ecological livability, the higher the degree of attention to the work of domestic waste separation and the better the infrastructure of waste separation. The more the village attaches importance to ecological and livable construction, the more the village attaches importance to domestic waste separation, and the better the infrastructure for waste separation, resulting in lower costs for residents to participate in domestic waste separation activities and more willingness to participate in domestic waste separation activities.

(3) External environment. The publicity means will positively and significantly affect the deviation of rural residents' willingness and behavior in waste separation at the 1% level, indicating that the greater the publicity efforts of villages, the lower the deviation of rural residents' willingness and behavior in waste separation, and the publicity efforts of villages often reflect the degree of importance villages attach to domestic waste separation; the more importance villages attach to domestic waste separation work, the more the publicity will affect the overall village appearance and form a good village culture, which will influence the villagers to participate in the separation of domestic waste. This suggests that the institutional rules to increase the cost of non-participation and the benefit of participation in waste separation are very effective in promoting participation and reducing the degree of divergence between willingness and behavior. On the whole, the change of external environment, through the change of policy mechanism and the strength of publicity, is very effective in improving villagers' willingness and behavior to participate in household waste separation, and the coefficients of both are the highest among all the influencing variables.

(4) Internal factors. The policy effect perception positively and significantly affects the paradox of rural residents' willingness to separate garbage and behavior at the 1% level, indicating that improving villagers' understanding of garbage separation and habitat improvement through publicity can effectively improve the situation of the paradox of behavior and willingness, and the higher the villagers' understanding of the significance of the behavioral work of domestic garbage separation, the more it can reduce the paradox of rural residents' willingness to separate garbage and behavior. The higher the level of villagers' understanding of the meaning of waste separation, the more it reduces the deviation of rural residents' intention and behavior. The perceived environmental behavior of villagers positively and significantly affects the divergence between the willingness and behavior of rural residents at the 1% level, indicating that there is a transmission effect

of the behavior. The effect of waste segregation is spread to those who do not want to participate in waste segregation activities.

### 3.2. ISM Analysis of the Factors Influencing the Paradox of Rural Residents' Willingness and Behavior to Separate Garbage

The above regression results show that the influential factors affecting rural residents' willingness to separate garbage and behavioral deviations are mainly eight factors: means of publicity, means of reward and punishment, perceived policy effects, villagers' perceptions of environmental behavior, gender, age, socioeconomic status, and ecological livability status. In this paper, we use $S_i(i = 1, 2, \ldots, 8)$, which denotes the above eight influencing factors, and $S_0$ denotes rural residents' willingness to separate garbage and behavioral deviation. Based on consultation with a total of 14 experts in the field of rural environmental management, the influence relationship between the factors is determined by combining the analysis of existing literature and relevant theories. It is assumed that if row factor i has influence on column factor j, it is denoted by V; if column factor j has influence on row factor i, it is denoted by A; if row factor i and column factor j have no influence relationship, it is denoted by O. The logical relationship of each influencing factor (see Figure 1 for details).

| A | A | A | A | A | A | A | A | S0 |
|---|---|---|---|---|---|---|---|---|
| V | O | O | O | V | V | O | S1 | |
| V | O | O | O | V | O | S2 | | |
| V | A | A | A | V | S3 | | | |
| V | A | A | A | S4 | | | | |
| O | O | O | S5 | | | | | |
| O | V | S6 | | | | | | |
| O | S7 | | | | | | | |
| S8 | | | | | | | | |

**Figure 1.** Logic diagram of influencing factors.

Referring to the logical relationship diagram of influencing factors, the adjacency matrix R of the influencing factors of rural residents' willingness to sort garbage and behavioral deviance can be obtained, and the reachable matrix M can be obtained by Boolean operation. Based on the Formula (4), the adjacent levels and the influencing factors of the same level in the reachable matrix M are further calculated and connected by using directed edges to obtain the hierarchical structure T for the influencing factors of rural residents' willingness to sort garbage and behavioral deviance (See Figure 2 for details).

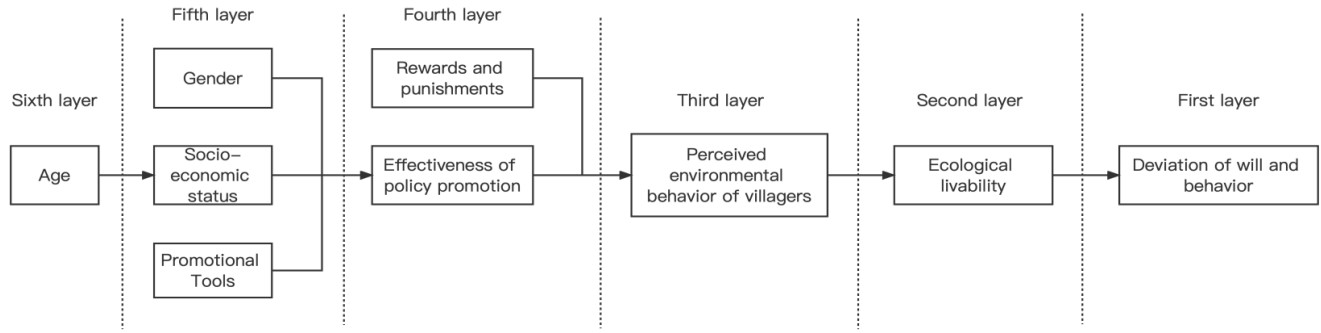

**Figure 2.** Interpretative Structural Modeling Model (ISM).

From Figure 2, it can be seen that ecological livability status is the surface-level direct factor that affects rural residents' willingness to separate garbage from behavioral deviance, reward and punishment means, and policy publicity effect, and villagers' environmental protection behavior perceptions are the intermediate level indirect factors that affect rural

residents' willingness to separate garbage from behavioral deviance; age, gender, socioeconomic status, and publicity means are the deep root issues that affect rural residents' willingness to separate garbage from behavioral deviance. There are four main transmission paths, which are: (1) "age → socioeconomic status → policy publicity effect → villagers' perception of environmental protection behavior → ecological livability → willingness and behavior paradox". (2) "gender → policy publicity effect → villagers' perception of environmental protection behavior → ecological livability → willingness and behavior paradox". (3) "Publicity → policy publicity effect → villagers' perception of environmental protection behavior → ecological livability → willingness and behavior paradox". (4) "Reward and punishment → villagers' perception of environmental protection behavior → ecological livability → willingness and behavior paradox ". The commonality of the four paths is that the effect of policy propaganda to enhance the perception of rural residents based on the villagers will observe the environmental behavior status of other villagers and examine the degree of attention to the construction of ecological livability; the degree of improvement in infrastructure in the village and its comprehensive impact on their own waste separation will be paradoxical to behavior. Among them, age, gender, socioeconomic status, and means of propaganda all influence the effect of policy propaganda, while the means of reward and punishment directly affect the perception of the environmental behavior of other villagers and the ecological livability of the village, thus influencing whether there is a deviation between willingness and behavior.

### 3.3. MICMAC Analysis of the Factors Influencing the Paradox of Rural Residents' Willingness and Behavior to Separate Garbage

The ISM model reveals the transmission paths and hierarchical structure between different factors, based on its inability to reflect the intensity of influence between hierarchical factors and transmission paths, the MICMAC analysis method is further adopted on the basis of the ISM model to calculate the dependence and magnitude of the driving force of each factor using the reachable matrix M and to analyze the intensity of influence of each factor and the effect of each factor on the overall system, all to make up for the shortcomings of the ISM model. Based on this, the dependency and driving force of each influencing factor are calculated using Equations (5) and (6), respectively (see Table 3 for details), and the graphs are made based on the dependency and driving force of each influencing factor (see Figure 3 for details).

**Table 3.** Dependence and driving force of the factors influencing the paradox of rural residents' willingness and behavior to separate waste.

|  | Dependency | Driving Force |
| --- | --- | --- |
| S0 | 9 | 1 |
| S1 | 1 | 5 |
| S2 | 1 | 4 |
| S3 | 5 | 4 |
| S4 | 7 | 3 |
| S5 | 1 | 5 |
| S6 | 1 | 6 |
| S7 | 2 | 5 |
| S8 | 8 | 2 |

Through MICMAC analysis, the 8 influencing factors can be divided into the following categories: Zone I represents the area of autonomous factors, which does not contain influencing factors; the dependence and driving force of the autonomous factors in this area are not strong and generally belong to independent factors, which have little correlation with other factors, simple relationship with other factors, and do not easily trigger a chain reaction. The dependency factors are generally strongly linked to other factors and easily controlled by other factors, but the driving force is not strong, and they are the most direct factors that constitute the deviation of rural residents' willingness and behavior to separate

garbage. Area III represents the area of associated factors, including the effect of policy propaganda (S3), associated factors have the characteristics of high dependency and high driving force, and generally belong to the transitional factors in the overall system, with the role of carrying on the top and bottom; Area IV represents the driving factors, including the means of propaganda (S1), rewards and punishments (S2), gender (S5), age (S6), and socioeconomic status (S7) driving factors of characterized by a high driving force and low dependence, not easily influenced, generally at the lower level of the ISM model, is the deepest factor affecting rural residents' willingness to sort garbage and behavioral paradoxes; Zone V is on the mean value line and is an adjustment factor, including villagers' environmental behavior cognition (S4), villagers' environmental behavior cognition is located between Zone II and Zone III, indicating that it has the dependence and association factors of dual characteristics: susceptible to other factors, while being a transitional factor in the overall system.

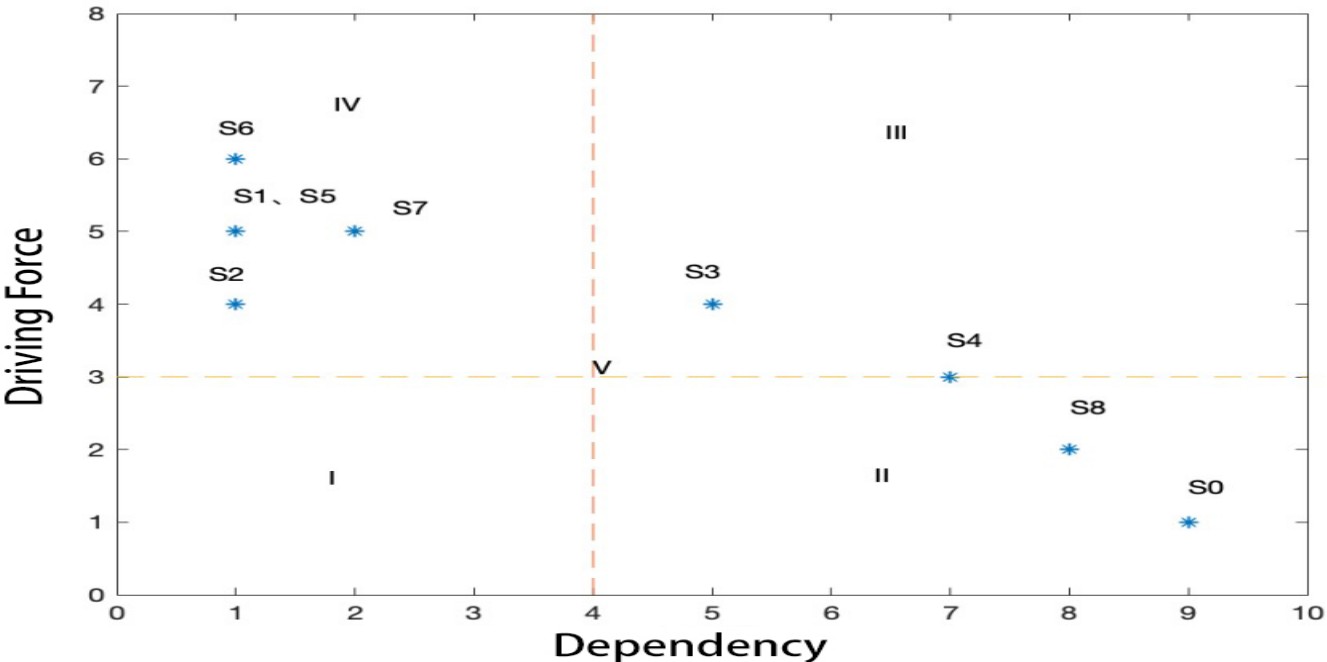

**Figure 3.** Dependency and driving force classification of the factors influencing rural residents' willingness to separate waste and behavioral paradoxes.

## 4. Discussion

The discrepancy between rural residents' willingness to separate garbage and their behavior indicates that establishing an effective mechanism for rural garbage separation still has a long way to go. In this paper, based on the survey data of CLES 2020 in rural areas of Jiangsu Province, we analyze the factors influencing the paradox of rural residents' willingness and behavior to separate garbage using the Logit model; we analyze the transmission path and hierarchical structure of the factors influencing the paradox of rural residents' willingness and behavior to separate garbage using the ISM model; finally, we further discuss the strength and effect of each influencing factor in the overall system using the MICMAC model. Finally, the MICMAC model was used to further discuss the strength and role of each influencing factor in the overall system. The main marginal contributions of this paper are as follows: (1) the research scope of the study on the paradox of willingness and behavior is enriched, and the paradox of willingness and behavior of waste separation can be compared with that of urban residents, and the similarities and differences of the paradox of willingness and behavior of waste separation between rural residents and urban villagers can be summarized. (2) Based on the existing research, the combined Logit-ISM model is expanded into a combined Logit-ISM-MICMAC model. The MICMAC model can

make up for the shortcomings of the ISM model and can further investigate the strength of influencing factors in the overall system and put forward corresponding countermeasures and suggestions in a more targeted manner. (3) Based on the variable selection of related studies, the village characteristics variables are included to improve the possible existence of willingness and behavior deviation in the overall system's influencing factors.

The results of this article are similar to those of Zuo Xiaofan [18], Jiang Lina [19], Liu Jiyao [20], Shen Xin [21], Cheng Huishan [22] and Stričík Michal [23]. The results of this article are similar to those of Zeng Qiyan, Chang Qian, and Wu Chunya. In comparison with the results of the existing studies on behavior-intention paradox, the results of the article are similar to those of Zeng Qiyan [24], Chang Qian [25] and Wu Chunya [26]. They all believe that cognitive factors, external environmental factors, individual economic factors, gender factors, and age factors affect the paradox of intention and behavior. In addition, by comparing the results of Chen Shaojun's [27] study, we found that: cognitive perception factors affect both urban and rural residents' willingness and behavior to separate garbage; environmental convenience and perfection (corresponding to ecological livability) also affect both urban and rural residents' willingness and behavior to separate garbage; for urban and rural residents, the reward and punishment mechanism is an important factor that affects the deviation of waste separation intention and behavior, but for urban residents, the publicity mechanism does not play a good role in reducing the deviation of intention and behavior, which may be due to the difference of educational resources between urban and rural residents. It can be seen that there are both commonalities and differences in the willingness and behavior of urban and rural residents to separate garbage. In future, we can discuss the differences in the factors influencing the willingness and behavior of urban and rural residents, so as to enrich the research perspectives and depth of the research on willingness and the behavior paradox.

Similarly, the study also has some limitations: (1) due to the limitations of the data, the study can only explore the paradoxical study of the willingness and behavior of rural residents in the Jiangsu Province, which cannot be properly compared and studied with the situation between different regions. (2) The responses of some variables in the study still receive the influence of subjective factors of the survey respondents, which may also lead to the subsequent ISM model and MICMAC model results. In order to control the influence of subjective factors on the results, the study used the results of the existing literature and the regression results of the existing data to synthesize the causal relationships among the influencing factors so as to mitigate the influence of subjectivity on the results on the basis of the traditional ISM model that allows experts to make judgments directly. Overall, the study is an enrichment and supplement to the existing related research, and the subsequent research still needs to improve the limitations and defects of the existing research.

## 5. Conclusions

Using data from 2204 questionnaires from CLES, the article analyzed the factors influencing the paradox of rural residents' willingness and behavior to separate garbage through a combined LOGIT-ISM-MICMAC model and came to the following conclusions.

(1) Through the regression analysis results, we found that the factors influencing rural residents' willingness to separate garbage and behavioral deviation are mainly eight factors: propaganda means, reward and punishment means, perceived policy effect, villagers' perception of environmental protection behavior, gender, age, socio-economic status, and ecological livability status. Among them, age and gender have a negative influence on rural residents' willingness to separate garbage and behavioral deviance, i.e., there are characteristics of an age difference and gender difference, while the remaining factors play a positive influence.

(2) The results of the ISM model found that age, gender, means of propagation, and socioeconomic status affect the effect of policy propagation and thus villagers' perceptions, and that rural residents with certain perceptions are influenced by other villagers' waste separation behaviors, i.e., whether rural residents' waste separation willingness and be-

havior are consistent is influenced by behavioral propagation, and on the basis of having behavioral propagation Villagers will consider the ecological and livable construction factors of the village, which in turn will influence whether the waste separation willingness and behavior deviate from each other. The reward and punishment mechanism directly affects the villagers' perceptions, which in turn affects whether the waste separation intention and behavior deviate from each other.

(3) The results of the MICMAC model showed that we should pay more attention to three factors: ecological livability, policy publicity effect, and villagers' environmental protection behavior cognition, and reduce the deviation of rural residents' waste separation behavior from their will by strengthening the ecological livability of villages, ensuring the policy publicity effect, and encouraging villagers' environmental protection behavior.

According to the above research findings, the following countermeasures are proposed: (1) Strengthen the environmental perception of rural residents, enhance publicity and education, and make rural residents recognize the importance and necessity of garbage classification work. (2) Encourage the garbage classification behavior of rural residents, formulate corresponding reward and punishment measures, especially to play the pioneering role of party members to play a behavior-spreading effect, and form a good social culture of garbage classification. (3) Strengthen the ecological and livable construction of villages, pay attention to village appearance, ensure the convenience of living, and improve infrastructure such as sewage and garbage treatment; strengthen the ecological livability of the village, focusing on the village appearance, guaranteeing the convenience of life, sewage and garbage management and other infrastructure improvements.

**Author Contributions:** Writing—original draft preparation, X.-Y.L.; funding acquisition and supervision, S.-W.H.; writing—review, Article Retouching and editing, Q.L.; Conceptualization, Q.-J.L.; data curation, Y.-S.Z.; All authors have read and agreed to the published version of the manuscript.

**Funding:** National Social Science Foundation of China General Project "Research on Modernization of Rural Environmental Governance System and Governance Capacity" (20BSH113).

**Institutional Review Board Statement:** Not applicable.

**Informed Consent Statement:** Not applicable.

**Data Availability Statement:** The link to access the data is as follows: https://jiard.njau.edu.cn/info/1033/1506.htm.

**Conflicts of Interest:** The authors declare no conflict of interest.

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
