# Peer review of "A Study on the Factors Influencing the Intention and Behavior Deviation of Rural Residents in Waste Separation—Based on LOGIT-ISM-MICMAC Combination Model"

_sustainability, doi:10.3390/su142215481_

Round 1

Reviewer 1 Report

Dear Authors:

 I have the following comments and questions: 

1.     Some parts are redundantly described. Please check them.

2.     L36-37: It is necessary to describe which country the authors would like to discuss.

3.     L68: Is “although social embedding” grammatically correct?

4.     L76: Is “...behavior The study…” OK?

5.     Equation (1): What is Pi?

6.     There is no definition of variables in Equation (4).

7.     What is mij in Equations (5) and (6)?

8.     Dependent variable in Table 1: The reviewer has a question for the definition of this variable. In this survey, are all residents assumed to be willing to act? If yes, how can the authors select only those who are willing to act? If no, those who answer “1” may include those who are not willing to act or behave. Is it OK to include such respondents with no willingness and no behavior?

9.     Perceived environmental behavior of villagers: The reviewer didn’t understand its definition.

10.  Social Networks in Table 1: What is the meaning of the standard deviation with more than 4.5 times of its averaged value? It seems to imply that a limited number of people have a huge number of mobile contacts as compared with the other people. Don’t you want to mention about these characteristics as well as a reason for introducing this variable as an independent variable.

11.  L239: Who are the experts? How many experts did the authors ask?

12.  Figure 1: This result is one of the most important analytical results in this study. What is the logic of deriving these results? How can the result of Figure 1 be validated? Without demonstrating them, results of Figure 1 seem arbitrary.

13.  Figure 1 (continued): Did the experts use results of correlation analyses of independent and dependent variables? Why didn’t the authors use the SEM (Structural Equation Modelling) by using the results of responses to begin with?

14.  Figures 2 & 3, Table 3: Given Figure 2, Dependency and Driving Force seem negatively correlated. The upper stream factors are located in Figure 2, the more Driving Force they are classified in. Honestly speaking, thus, the reviewer did not find strong additional new values in Table 3 and Figure 3.

15.  L316: Is “the structural equation model” correct?

16.  3. Discussions: The reviewer did not clearly understand the originality of this work.

17.  L389-394: The reviewer did not understand reason(s) for this description.

Author Response

We would like to thank the reviewers for their comments on the article, and will now revise and improve the article in response to the reviewers' comments as follows:

Point 1: Some parts are redundantly described. Please check them.

Response 1: Translation professionals have been asked to check and improve the phrasing of the article.

Point 2: L36-37: It is necessary to describe which country the authors would like to discuss.

Response 2: The country has been added to the sentence L36-37 as requested by the reviewer, which was amended to read: In China's Five-Year Action Plan for the Improvement and Upgrading of Rural Habitat (2021-2025), it is also mentioned that "we should speed up the source classification and reduction of rural domestic waste, actively explore ways to meet the characteristics of rural areas and farmers' habits, and Simple and easy to implement the classification and treatment model, reduce the amount of garbage out of the village treatment."

Translated with www.DeepL.com/Translator (free version)

Point 3: L68: Is “although social embedding” grammatically correct?

Response 3:Translation professionals have been asked to check and improve the phrasing of the article.

Point 4:  L76: Is “...behavior The study…” OK?

Response 4: Translation professionals have been asked to check and improve the phrasing of the article.

Point 5:  Equation (1): What is Pi?

Response 5: Pi represents the probability, where the probability of Yi taking 1 is Pi, and the probability of Yi taking 0 is 1-Pi. The original text has been revised according to the reviewer's comments, and is modified as follows: Eq. , denotes the paradox of rural residents' willingness to separate garbage from their behavior, where the probability of Yi taking 1 is Pi, and the probability of Yi taking 0 is 1-Pi. denotes the possible factors that may affect the paradox of rural residents' willingness to separate garbage from their behavior , denotes the regression coefficient of the nth independent variable, and denotes the random error term.

Translated with www.DeepL.com/Translator (free version)

Point 6:  There is no definition of variables in Equation (4).

Response 6: The variables have been described for Equation 4, which is added as follows: where  is the reachable set, which is the set consisting of all the columns with matrix element 1 in the row corresponding to the influence element  in the reachable matrix M;  is the prior set, which is the set consisting of all the rows with matrix element 1 in the column corresponding to the influence element  in the reachable matrix M, whose intersection is defined as . When and only when , all the elements in  corresponding to of the influence elements are the top elements, and then the top elements are eliminated and this step is continued until the bottom level.

Point 7:  What is mij in Equations (5) and (6)?

Response 7: The description of the mij variable has been illustrated below Equation 5 and Equation 6 with the meaning of the influences in the reachable matrix M.

Point 8: Dependent variable in Table 1: The reviewer has a question for the definition of this variable. In this survey, are all residents assumed to be willing to act? If yes, how can the authors select only those who are willing to act? If no, those who answer “1” may include those who are not willing to act or behave. Is it OK to include such respondents with no willingness and no behavior?

Response 8: The treatment of the dependent variable in the article is based on two questions: "Are you willing to separate household waste? (1=yes; 0=no)" and "Are you willing to separate your household waste? (1=yes; 0=no)" and defined the existence of paradox for the group with willingness without behavior and without willingness with behavior among them, and for the group with willingness with behavior and without willingness without behavior defined as no paradox. Additional explanations have been provided in Table 1 of the article.

Point 9:  Perceived environmental behavior of villagers: The reviewer didn’t understand its definition.

Response 9: Villagers' perceptions of environmental behavior are positioned as villagers' perceptions of other villagers' environmental behavior, and there may be a herd effect in behavioral research, i.e., the behavior of others may have an impact on individual willingness or behavior, so villagers' perceptions of environmental behavior are included, mainly to measure the survey respondents' attitudes and perceptions of others' environmental behavior, and to use them as internal variables to measure whether they may have an impact on the deviation of willingness and behavior.

Point 10: Social Networks in Table 1: What is the meaning of the standard deviation with more than 4.5 times of its averaged value? It seems to imply that a limited number of people have a huge number of mobile contacts as compared with the other people. Don’t you want to mention about these characteristics as well as a reason for introducing this variable as an independent variable.

Response 10: In existing studies, social network status is mostly measured by more subjective variables such as contact frequency, but the number of cell phone contacts can be a more objective measure of social network status. At present, China is facing aging and hollowing out of the countryside, which leads to the fact that most of the elderly and children left behind in rural China have fewer cell phone contacts, while a small number of people have a large number of contacts for two main reasons: First, local rural cadres are the core contacts of the village network, and villagers with cadre status will have more contacts. Second, young people who go out to work will also have a higher number of contacts. However, these two groups are fewer in number in rural areas, and most young people tend to return home only during major festivals, which leads to the problem of huge standard deviation, but what can be ensured is that the CLES data is a publicly available data, a rural research led by Nanjing Agricultural University, and the reliability of its data can be guaranteed.

Point 11: L239: Who are the experts? How many experts did the authors ask?

Response 11: In addition to the authors Huang Senwei and Lu Qiujia, there are mainly Su Shipeng, Wei Yuanzhu, Du Yanqiang, Shi Wei, Liu Yan, Wang Yanyan and other masters, associate professors and professors who mainly study rural economy and rural governance, a total of 14 experts and scholars. According to the reviewers' comments, the description of the experts in the text was revised to be based on consulting a total of 14 experts in the field of rural environmental governance. It should be added that the number of experts consulted in ISM models in existing ISM model-related studies is between 10-20, and the number and quality of experts consulted in this study meet the basic requirements of ISM model research.

Point 12:  Figure 1: This result is one of the most important analytical results in this study. What is the logic of deriving these results? How can the result of Figure 1 be validated? Without demonstrating them, results of Figure 1 seem arbitrary.

Response 12: According to the traditional ISM model, the determination of the influence relationship between the influencing factors is often based on the judgment of expert scholars, but this can lead to the problem of subjectivity. Therefore, the study combined the judgments of 12 experts and scholars with relevant findings from existing studies and the regression results using existing data to arrive at the results in Figure 1.

Point 13: Figure 1 (continued): Did the experts use results of correlation analyses of independent and dependent variables? Why didn’t the authors use the SEM (Structural Equation Modelling) by using the results of responses to begin with?

Response 13: According to the traditional ISM model, the determination of the influence relationship between influencing factors is often based on the judgment of expert scholars, but this can present the problem of subjectivity. In this regard, the article combines data to have a causal relationship between variables and combines the results of existing studies (listed in the references) to assist in verifying expert judgment. The main reasons for not using SEM models are: 1. data limitations, the CLES questionnaire data does not support the construction of SEM models. 2. although SEM models can test the mediating effects, they cannot analyze the influence paths of all influencing factors, and can only verify the existence of some influence paths.

Point 14: Figures 2 & 3, Table 3: Given Figure 2, Dependency and Driving Force seem negatively correlated. The upper stream factors are located in Figure 2, the more Driving Force they are classified in. Honestly speaking, thus, the reviewer did not find strong additional new values in Table 3 and Figure 3.

Response 14: MICMAC classifies indicators into correlation factors, adjustment factors, drivers, dependence factors and autonomous factors based on their roles, which can be expressed visually through the directional lines of the skeleton diagram. The autonomy factor has fewer lines pointing to and deriving from it, and its correlation ranking is lower; the dependency factor is mostly a pointing line, and the driver is mostly a deriving line; the correlation factor and the adjustment factor have more total lines, and the difference is that the adjustment factor is a factor whose driving force or dependency is at the mean level, and it is a key factor linking adjacent layers or between layers. core factors and have an adjusting effect. The factors at the bottom of the hierarchy have an adjusting effect because they are directly related to the core factors, while the associated factors focus more on the degree of linkage with other factors, and require higher driving force and dependency to reach above the mean level. The model and method do not lie in considering the linear relationship between the influencing factors, but secondly, measuring the importance of the influencing factors in the system through the two elements of dependency and driving force, so that the most important countermeasure suggestions can be realized in a targeted manner.

Point 15:  L316: Is “the structural equation model” correct?

Response 15: Had the article checked and improved by translation professionals. Changed "the structural equation model" to ISM model.

Point 16:  3. Discussions: The reviewer did not clearly understand the originality of this work.

Response 16: The main marginal contributions of the article are as follows: 1) It enriches the scope of the study on the paradox of willingness and behavior, and can compare the paradox of willingness and behavior of waste separation with that of urban residents, and summarize the similarities and differences of the paradox of willingness and behavior of waste separation between rural residents and urban villagers. 2) Based on the existing research, the combined Logit-ISM model is extended to the combined Logit-ISM-MICM model. -The MICMAC model can make up for the shortcomings of the ISM model, further investigate the strength of the influencing factors in the overall system, and put forward corresponding countermeasures and suggestions in a more targeted manner. 3) Based on the variable selection of related studies, the village characteristics variables are included, which can more fully improve the possible influences of the overall system of willingness and behavior deviation. Factors.

Point 17: L389-394: The reviewer did not understand reason(s) for this description.

Response 17: The conclusion section mainly provides a review and summary of the results of the Logit-Ism-Micmac model. Among them, parts L389-394 are for the summary of the results of the Logit regression model.

Reviewer 2 Report

- Table 1: The table is difficult to read, I suggest using dividing lines.

- "Research methods, data sources and variable selection" and Result and Analysis sections are clear, detailed, but I think if you could describe the current state of waste management it would help us to understand the results.

- I suggest that the references in the literature should be extended with international examples, especially in the field of waste management. You have used few references in your work.

- Has the application of the LOGIT-ISM-MICMAC combination model brought new results for waste management? I recommend you expand this section. 

Author Response

We would like to thank the reviewers for their comments on the article, and will now revise and improve the article in response to the reviewers' comments as follows:

Point 1: Table 1: The table is difficult to read, I suggest using dividing lines.

Response 1: A split line has been added to the Table 1 section as requested by the reviewer.

Point 2: "Research methods, data sources and variable selection" and Result and Analysis sections are clear, detailed, but I think if you could describe the current state of waste management it would help us to understand the results.

Response 2: To address the current situation of waste treatment, the article has cited relevant studies and briefly described them in the introduction section, the main content is: At present, there are various rural household waste treatment models in China, but the efficiency of different rural household waste treatment models has not been greatly improved. How to stimulate the vitality and improve the efficiency of rural domestic waste treatment models? It has been pointed out that to be efficient, the government's regulatory role and the participation of rural residents must be strengthened. However, on the whole, the participation level of rural residents is generally low, and there is a discrepancy between willingness to participate and participation behavior, mainly showing the characteristics of "high willingness-low behavior". In this regard, it is of practical and theoretical significance to study how to promote the transformation from willingness to behavior and to study the reasons and influencing factors of the deviation between willingness and participation behavior to promote the efficiency of rural domestic waste management model, which is also the key to promote rural domestic waste management.

Point 3: I suggest that the references in the literature should be extended with international examples, especially in the field of waste management. You have used few references in your work.

Response 3: The introduction has been supplemented with additional citations to the relevant international literature on the current status of research on domestic waste management as requested by the reviewers.

The additions are as follows: studies on waste separation in developing countries show that household economic and social welfare, the amount and distance of infrastructure for waste disposal already neighborhood factors are important factors influencing the willingness and behavior of waste separation. For developed countries, empirical studies in Taiwan and Japan have shown that incentives and coercive measures are important factors influencing the willingness and behavior of waste separation. There are also studies by examining the factors influencing household recycling behavior in Western European countries and comparing them with household recycling behavior in the U.S. The results show that socio-economic factors and incentive policies affect household recycling behavior, and household recycling behavior in Western European countries is more influenced by incentive policies, and although Americans' recycling behavior is also influenced by policy factors, in comparison with Western European countries, household recycling behavior is more influenced by socioeconomic conditions and infrastructure than in Western European countries.

The following references were added.

  1. Adzawla W, Tahidu A, Mustapha S, Azumah SB. do socioeconomic factors influence households' solid waste disposal systems? from Ghana. waste Management & Research. 2019;37(1_suppl):51-57.
  2. Tewodros Tadesse, Arjan Ruijs, Fitsum Hagos,Household waste disposal in Mekelle city, Northern Ethiopia,Waste Management,,2008,28(10):2003- 2012,
  3. Zhao, L.; Chen, H. Exploring the Effect of Family Life and Neighbourhood on the Willingness of Household Waste Sorting. sustainability 2021, 13, 13653.
  4. Kuo, YL., Perrings, C. Wasting Time? Recycling Incentives in Urban Taiwan and Japan. 437.
  5. Yu-Long Chao, Time series analysis of the effects of refuse collection on recycling: Taiwan's
  6. Dai X, Han Y, Zhang X, et al. Comparison between students and residents on determinants of willingness to separate waste and waste separation behaviour in Zhengzhou, China. waste Management & Research. 2017;35(9):949-957.
  7. Kipperberg, G. A Comparison of Household Recycling Behaviors in Norway and the United States. environment Resource Economics,2007,36, 215 Environmental Resource Economics, 2007, 36, 215 -235.
  8. Cheng Huishan, Rui Quanquan, Yu Kunyong, Li Xiaohe, Liu Jian,Exploring the Influencing Paths of Villagers' Participation in the Creation of Micro-Landscapes: An Integrative Model of Theory of Planned Behavior and Norm Activation Theory, Frontiers in Psychology, 2022(13).  
  9. Shen Xin,Chen Bowei,Leibrecht Markus,Du Huanzheng. The Moderating Effect of Perceived Policy Effectiveness in Residents’ Waste Classification Intentions: A Study of Bengbu, China[J]. Sustainability,2022,14(2).
  10. Stričík Michal,Čonková Monika. Key Determinants of Municipal Waste Sorting in Slovakia[J]. Sustainability,2021,13(24).

Point 4: Has the application of the LOGIT-ISM-MICMAC combination model brought new results for waste management? I recommend you expand this section. 

Response 4: The LOGIT-ISM-MICMAC model mainly analyzes those factors that affect willingness and behavioral deviance first?The ISM model is to sort out the conduction paths of significant influencing factors, and the MICMAC model mainly analyzes the importance of influencing factors in the system.By using a combination of these three models, we have different findings than previous studies: (1) The ISM model shows that there are four main transmission paths, namely: 1) "age → socioeconomic status → policy advocacy effect → villagers' perception of environmental behavior → ecological livability status → willingness and behavioral deviance". 2) "gender → policy advocacy effect → villagers' perception of environmental behavior → ecological livability status → willingness and behavioral deviance". 3) "gender → policy advocacy effect → villagers' perception of environmental behavior → ecological livability (2) "gender → policy publicity effect → villagers' perception of environmental protection behavior → ecological habitability → willingness and behavioral dissonance". 3) "publicity means → policy publicity effect → villagers' perception of environmental protection behavior → ecological habitability → willingness and behavioral dissonance". 4) "reward and punishment means → villagers' perception of environmental protection behavior → ecological habitability → willingness and behavioral dissonance". The commonality of the four paths lies in the policy and practice. The commonality of the four paths is that on the basis of the policy propaganda effect to improve the perception of rural residents, villagers will observe the environmental behavior status of other villagers and examine the degree of attention to the construction of ecological livability and the degree of improvement of infrastructure in the village combined to influence whether their own willingness to separate garbage and behavior will produce deviations. Among them, age, gender, socioeconomic status, and means of propaganda will affect the effect of policy propaganda, while the means of reward and punishment will directly affect the perception of environmental behavior of other villagers and the ecological livability of the village, thus affecting whether the intention and behavior will be deviated from each other. (2) The results of MICMAC model show that we should pay more attention to three factors: ecological habitability, policy propaganda effect, and villagers' perception of environmental protection behavior, and reduce the deviation between rural residents' behavior and willingness to separate garbage by strengthening the ecological habitability of villages, ensuring policy propaganda effect, and encouraging villagers' environmental protection behavior. All of the above is reflected in the article.

Reviewer 3 Report

This is an interesting paper and I enjoyed reading. There are few points to increase the quality of the paper.

1.     The paper lacks review. It will be interesting to know the factors that influence the rural resident’s behavior around the world in waste separation. They must be added as a part of introduction or as a separate section.

2.     Mention limitations of the present study and scope for future research in conclusion. 

Author Response

We would like to thank the reviewers for their comments on the article, and will now revise and improve the article in response to the reviewers' comments as follows:

Point 1: The paper lacks review. It will be interesting to know the factors that influence the rural resident’s behavior around the world in waste separation. They must be added as a part of introduction or as a separate section.

Response 1: The introduction has been supplemented with additional citations to the relevant international literature on the current status of research on domestic waste management as requested by the reviewers.

The additions are as follows: studies on waste separation in developing countries show that household economic and social welfare, the amount and distance of infrastructure for waste disposal already neighborhood factors are important factors influencing the willingness and behavior of waste separation. For developed countries, empirical studies in Taiwan and Japan have shown that incentives and coercive measures are important factors influencing the willingness and behavior of waste separation. There are also studies by examining the factors influencing household recycling behavior in Western European countries and comparing them with household recycling behavior in the U.S. The results show that socio-economic factors and incentive policies affect household recycling behavior, and household recycling behavior in Western European countries is more influenced by incentive policies, and although Americans' recycling behavior is also influenced by policy factors, in comparison with Western European countries, household recycling behavior is more influenced by socioeconomic conditions and infrastructure than in Western European countries.

The following references were added.

  1. Adzawla W, Tahidu A, Mustapha S, Azumah SB. do socioeconomic factors influence households' solid waste disposal systems? from Ghana. waste Management & Research. 2019;37(1_suppl):51-57.
  2. Tewodros Tadesse, Arjan Ruijs, Fitsum Hagos,Household waste disposal in Mekelle city, Northern Ethiopia,Waste Management,,2008,28(10):2003- 2012,
  3. Zhao, L.; Chen, H. Exploring the Effect of Family Life and Neighbourhood on the Willingness of Household Waste Sorting. sustainability 2021, 13, 13653.
  4. Kuo, YL., Perrings, C. Wasting Time? Recycling Incentives in Urban Taiwan and Japan. 437.
  5. Yu-Long Chao, Time series analysis of the effects of refuse collection on recycling: Taiwan's
  6. Dai X, Han Y, Zhang X, et al. Comparison between students and residents on determinants of willingness to separate waste and waste separation behaviour in Zhengzhou, China. waste Management & Research. 2017;35(9):949-957.
  7. Kipperberg, G. A Comparison of Household Recycling Behaviors in Norway and the United States. environment Resource Economics,2007,36, 215 Environmental Resource Economics, 2007, 36, 215 -235.
  8. Cheng Huishan, Rui Quanquan, Yu Kunyong, Li Xiaohe, Liu Jian,Exploring the Influencing Paths of Villagers' Participation in the Creation of Micro-Landscapes: An Integrative Model of Theory of Planned Behavior and Norm Activation Theory, Frontiers in Psychology, 2022(13).  
  9. Shen Xin,Chen Bowei,Leibrecht Markus,Du Huanzheng. The Moderating Effect of Perceived Policy Effectiveness in Residents’ Waste Classification Intentions: A Study of Bengbu, China[J]. Sustainability,2022,14(2).
  10. Stričík Michal,Čonková Monika. Key Determinants of Municipal Waste Sorting in Slovakia[J]. Sustainability,2021,13(24).

Point 2: Mention limitations of the present study and scope for future research in conclusion. 

Response 2: The direction and scope of future research has been explored and proposed in the second paragraph of the discussion in the article, and the research limitations have been added in the discussion section according to the reviewer's opinion, with the following main additions: Again, there are certain limitations of the study: 1) Subject to the limitations of the data, the study can only explore the paradoxical study of the willingness and behavior of the rural residents in Jiangsu province to separate garbage, which cannot be well compared with the situation between different regions It is not possible to compare and study the situation between different regions.2) The responses of some variables in the study are still influenced by subjective factors of the respondents, which may lead to inaccurate results of the subsequent ISM model and MICMAC model. In order to control the influence of subjective factors on the results, the study uses the results of the existing literature and the regression results of the existing data to synthesize the results of the traditional ISM model on the basis of direct expert judgment. In order to control the influence of subjectivity on the results, the study uses the results of the existing literature and the regression results of the available data to synthesize the causal relationships among the influencing factors, so as to reduce the influence of subjectivity on the results. Overall, the study is an enrichment and supplement to the existing related studies, and the subsequent studies still need to improve the limitations and defects of the existing studies.

Round 2

Reviewer 1 Report

Dear Authors:

Thank you for your revisions.

The reviewer still has some questions and comments.

1)     Equation (1): Please describe the definition of Pi in the manuscript.

2)     Line 117: Is the expression of “explanatory structural model (ISM)” OK?

3)     Equation (5) and (6): Please describe the definition of E & F in the manuscript. Shouldn’t Ei be Ej?

4)     Table 1. Dependent Variable: The reviewer has two questions.

a)      The authors interpret “Are you willing to separate your household waste?” as a question to measure the behavior.” Is this interpretation reasonable?

b)     The authors define paradox as an answer with one willing and one unwilling regardless of type of question. Is this definition OK? It seems relevant to define paradox as an answer with willing to separate household waste but unwilling to separate your household waste.

Thank you.

Author Response

We would like to thank the reviewers for their comments on the article, and will now revise and improve the article in response to the reviewers' comments as follows:

Point 1: Equation (1): Please describe the definition of Pi in the manuscript.

Response 1:Pi has been added and explained.The specific additions are as follows:In the equation(1), it denotes the paradox of rural residents' willingness to separate garbage from their behavior, where the probability of Yi taking 1 is  and the probability of Yi taking 0 is .

Point 2: Line 117: Is the expression of “explanatory structural model (ISM)” OK?

Response 2: After the authors reviewed the relevant research, they found that explanatory structural model is used by some people for this expression, but in fact, it would be more appropriate to use Interpretative Structural Modeling (ISM) Model. Therefore, the authors have replaced explanatory structural model with Interpretative Structural Modeling Model where it appears in the article.

Point 3: L68:  Equation (5) and (6): Please describe the definition of E & F in the manuscript. Shouldn’t Ei be Ej?

Response 3:Thanks to the reviewers for their careful review of the article, and after comparing with existing research formulas, we found that Ei should indeed be changed to Ej.The definitions of Ej and Fi have now been added and Ei has been revised to Ej.

Point 4:  Table 1. Dependent Variable: The reviewer has two questions.

a) The authors interpret “Are you willing to separate your household waste?” as a question to measure the behavior.” Is this interpretation reasonable?

b)The authors define paradox as an answer with one willing and one unwilling regardless of type of question. Is this definition OK? It seems relevant to define paradox as an answer with willing to separate household waste but unwilling to separate your household waste.

Response 4:

a) First, the measurement behavior was measured using the question "Do you separate your household waste? (1=yes; 0=no)", and the article has a duplicate error, which has been revised.

b) The questions used in the article were "Would you like to sort your household waste? (1=yes; 0=no)" to measure willingness, and "Do you sort your household waste? (1=yes; 0=no)" to measure behavior. The groups with willingness without behavior and without willingness with behavior were defined as contradictory, and the groups with willingness with behavior and without willingness without behavior were defined as non-contradictory. This treatment of expressing willingness and behavior by two dichotomous variables and measuring willingness-behavior contradiction by these two variables is also used in related articles. The reason for the reviewer's question should be the repetition of the wrong expression in the text, which was originally expressed through the question "Would you like to sort household waste?" This question was expressed as willingness and behavior. (1=yes; 0=no)", which was actually caused by the author's translation error. The article was revised as follows: Do intentions and behaviors contradict each other?0=contradictory; 1=no contradictory. Use the question "Would you like to separate your household garbage? (1=Yes; 0=No)" and "Do you separate your household garbage? (1=yes; 0=no)", the groups with and without willingness were defined as having contradiction and the groups with and without willingness were defined as having no contradiction.
